# Merging Text Transformer Models
# from Different Initializations

**Neha Verma**                                                                          *nverma7@jhu.edu*
*Center for Language and Speech Processing, Johns Hopkins University*

**Maha Elbayad**                                                                        *elbayadm@meta.com*
*Meta*

**Reviewed on OpenReview:** `https://openreview.net/forum?id=nWnYSLncXa`

## Abstract

Recent work on permutation-based model merging has shown impressive low- or zero-barrier mode connectivity between models from completely different initializations. However, this line of work has not yet extended to the Transformer architecture, despite its dominant popularity in the language domain. Therefore, in this work, we investigate the extent to which separate Transformer minima learn similar features, and propose a model merging technique to investigate the relationship between these minima in the loss landscape. The specifics of the architecture, like its residual connections, multi-headed attention, and discrete, sequential input, require specific interventions in order to compute model permutations that remain within the same functional equivalence class. In merging these models with our method, we consistently find lower loss barriers between minima compared to model averaging, across models trained on a masked-language modeling task or fine-tuned on a language understanding benchmark. Our results show that the minima of these models are less sharp and isolated than previously understood, and provide a basis for future work on merging separately trained Transformer models.

## 1 Introduction

The geometry of loss landscapes is the subject of extensive prior work attempting to understand the behavior and properties of different minima (Kawaguchi, 2016; Li et al., 2018; Du et al., 2019). Prior work on understanding the loss landscapes of deep neural networks has found different types of geometric paths of low loss between converged models, demonstrating a degree of connectedness between these separately trained minima (Freeman & Bruna, 2017; Garipov et al., 2018; Tatro et al., 2020). These findings have restructured our understanding of the relationship between different minima in loss space by uncovering entire low-loss regions in which several minima may be co-located.

A related line of work finds specifically *linear* paths between differently initialized models, where the linear interpolations of these minima have low loss like either endpoints (Entezari et al., 2022; Ainsworth et al., 2023). This connectivity is feasible by transforming the weights of the models, without changing their function, in order to compare them within a more similar loss space. The types of transformations exploit inherent model symmetries that are achievable via permuting model features, where the same underlying function may have numerous valid parameter configurations. So far, this work has mostly covered simple architectures like multi-layer perceptrons or VGGNets (Simonyan & Zisserman, 2015). Some work has included ResNets as well (He et al., 2016), and show that only wider ResNets can be merged with much smaller loss than standard versions (Ainsworth et al., 2023; Stoica et al., 2024).

While this research has led to important conclusions about loss landscapes between separately trained models, the Transformer architecture has been largely unexplored in terms of understanding its permutation symmetries and loss landscape geometry. Prior work has emphasized the importance of understanding loss landscape geometry, as a better understanding of loss geometry can lead to improvements in optimization techniques, ensembling, and model merging techniques (Garipov et al., 2018; Ainsworth et al., 2023). There has been some prior work in merging Transformer architectures (Jin et al., 2023; Imfeld et al., 2023), but these either do not account for models from separate initializations,

or do not create valid permutation mappings that transform a model into another member of its symmetric equivalence group (Chen et al., 1993).

Therefore, in this work, we explore the extent to which separately trained Transformer models learn similar representations, and then propose permutation-based merging method to align representations from these separate models. With this, we can study the extent of the connectivity between their minima in loss space in order to extend our understanding of loss landscape geometry to this important and popular architecture. We specifically investigate their connectivity through the lens of permutation-invariant linear mode connectivity (Entezari et al., 2022). Our permutation-based merging method builds upon prior work from Ainsworth et al. (2023), but focuses on necessary interventions needed for the Transformer architecture, such as merging Multi-Headed Attention, and the inputs/outputs of each residual connection.

Our contributions are the following:

1. We introduce a new model merging algorithm based on model permutations that combines Transformers from separate initializations.

2. We demonstrate reduced loss barriers between masked language models trained from completely separate initializations compared to vanilla merging.

3. We extend our approach to fine-tuned models and show consistently smaller loss barriers between models compared to vanilla merging.

In Section 2, we discuss related work to place our contributions in context of the existing literature. In Section 3, we describe our method for computing the permutations needed to align two Transformer minima in order to compare them in loss space. We outline the models, data, and evaluation used to test our approach in Section 4. Finally, we discuss our findings and analysis in Section 5.[1]

## 2   Related Work

**Loss Landscape & Mode Connectivity.**   Deep neural networks are typically trained by optimizing a loss function with an SGD variant. Loss landscapes of these networks have been shown to contain infinitely many global minimizers that are equally reachable via SGD (Kawaguchi, 2016; Du et al., 2019). Overparameterization is one of the reasons behind the abundance of minima leading to different functions that behave similarly on the training data (Nguyen et al., 2019; Simsek et al., 2021; Liu et al., 2022). Permutation and scaling invariances also lead to functionally identical minima that differ in the weight space (Tatro et al., 2020; Entezari et al., 2022).

Prior work has established that the optima of loss functions are in fact connected by simple curves over which training and test accuracy are nearly constant (no loss barrier) (Freeman & Bruna, 2017; Garipov et al., 2018; Draxler et al., 2018). This phenomenon is referred to as *mode connectivity*. Entezari et al. (2022) conjectured that if the permutation invariances of neural networks are taken into account, these optima are linearly mode connected, i.e. the linear path connecting these two models has no loss barrier. This is *linear mode connectivity*.

**Interpolating Models**   Empirically, linear interpolation between neural network weights has become an important tool. In the context of fine-tuning the same large pre-trained model, averaging models enabled state-of the art accuracy on ImageNet (Wortsman et al., 2022). Wortsman et al. (2022); Rame et al. (2022) established that if fine-tuned models lie in a single low error basin, then weight averaging performs similarly to ensembling. It is however not guaranteed that finetuned models (starting from the same initialization) will reside in the same loss basin.

Prior work on linear interpolation-based model merging has focused on improving the algorithms used to bring the hidden units of two networks into alignment, in order to reduce the barrier to interpolation between them. Singh & Jaggi (2020) develop a strong optimal transport-based method which allows linear interpolation between a pair of ResNet networks (He et al., 2016). Entezari et al. (2022) use an approach based on simulated annealing (Zhan et al., 2016) in order to find permutations such that wide MLPs trained on MNIST can be linearly interpolated with a barrier of nearly zero. Ainsworth et al. (2023) develop several permutation-based algorithms for MLPs and ResNets, and demonstrate

---

[1]We release our code at `https://github.com/nverma1/merging-text-transformers`

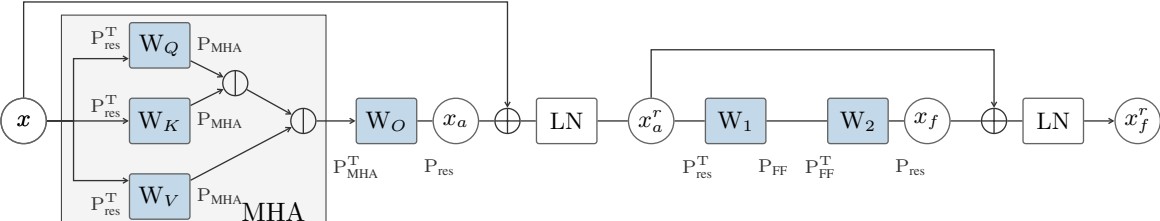

Figure 1: Example of a Transformer layer with its parameters outlined in blue boxes, specific hidden states as circles, and the flow of operations indicated with arrows. ⬭ indicates the dot-product operation, and ⊕ indicates addition. LN refers to LayerNorm modules. We include permutation and inverse permutation matrices at each weight matrix to indicate the proposed operations from our method. $\boldsymbol{P}_{\text{res}}$ refers to the residual permutation, $\boldsymbol{P}_{\text{MHA}}$ refers to the multi-headed attention (MHA) permutation, and $\boldsymbol{P}_{\text{FF}}$ aligns the feed-forward layers.

zero-barrier linear mode connectivity on widened ResNets. Stoica et al. (2024) extend this work and allow for feature merges to happen within each model as well, removing internal feature redundancies and improving model merging outcomes.

## 3  Proposed Transformer Merging Method

In this section, we describe the components of our method that address how to permute specific portions of a Transformer model $\boldsymbol{\theta}_B$ in order to bring them into alignment with a separately trained model, $\boldsymbol{\theta}_A$. Extending permutation-based model merging to Transformers is non-trivial as they have more complicated connections than simpler MLP-like architectures. We discuss how we find permutation matrices computed from feature correlations, and then we specifically address the parameters involved in multi-headed attention, residual connections, and pre- and post-net parameters. We diagram a Transformer layer in Figure 1, and the proposed permutation operations that we describe in detail throughout this section.

### 3.1  Computing Correlation and Permutation Matrices

Given two models trained on the same data but from from separate initializations, namely $\boldsymbol{\theta}_A$ and $\boldsymbol{\theta}_B$, we compute post-activation features for each layer or sublayer parameter $\boldsymbol{W}_\ell \subset \boldsymbol{\theta}$ in order to determine which features might correspond between models (Ainsworth et al., 2023; Stoica et al., 2024). In our setting, features are computed at the token level, and all special non-vocabulary and non-padding tokens are always included (such as [SEP], [CLS]). At a given layer or sublayer, we compute $d$-dimensional activations across $n$ tokens from both models $\boldsymbol{X}_A, \boldsymbol{X}_B \in \mathbb{R}^{n \times d}$, and then determine feature relatedness via cross-correlation computed as the following:

$$\boldsymbol{C} = \text{corr}(\boldsymbol{X}_A, \boldsymbol{X}_B) = \frac{\mathbb{E}[(\boldsymbol{X}_A - \boldsymbol{\mu}_{\boldsymbol{X}_A})^{\mathrm{T}}(\boldsymbol{X}_B - \boldsymbol{\mu}_{\boldsymbol{X}_B})]}{\boldsymbol{\sigma}_{\boldsymbol{X}_A}\boldsymbol{\sigma}_{\boldsymbol{X}_B}}, \tag{1}$$

where $\boldsymbol{\sigma}$ are feature standard deviations, and $\boldsymbol{\mu}$ are feature means. We choose to standardize and mean-center the features before comparing them because the magnitude of feature values can vary greatly in some pre-trained text Transformers (Puccetti et al., 2022). Then, given feature cross-correlations $\boldsymbol{C} \in \mathbb{R}^{d \times d}$, we compute the optimal permutation as the following:

$$\pi^* = \arg\max_\pi \sum_{i=1}^{d} \boldsymbol{C}_{i,\pi(i)} \tag{2}$$

where $\pi : \{1, 2, ..., d\} \rightarrow \{1, 2, ..., d\}$ is a permutation mapping. This optimization problem finds the feature correspondences between the two models that lead to the highest total correlation captured. This problem an instance of the assignment problem and can be solved using the Jonker-Volgenant algorithm (Crouse, 2016; Tatro et al., 2020; Ainsworth et al., 2023).

After converting the map $\pi^*$ to its corresponding permutation matrix $\boldsymbol{P}$, we can apply $\boldsymbol{P}$ to the original weight matrix $\boldsymbol{W}_\ell^B \subset \boldsymbol{\theta}_B$ so that the order of the layer's features most closely resembles that of $\boldsymbol{W}_\ell^A \subset \boldsymbol{\theta}_A$:

$$\boldsymbol{W}_\ell^{B'} \leftarrow \boldsymbol{P}\boldsymbol{W}_\ell^B. \tag{3}$$

We then apply $\boldsymbol{P}^{\mathrm{T}} = \boldsymbol{P}^{-1}$ to the next layer in order to unpermute the new ordering in model $\theta_B$:

$$\boldsymbol{W}_{\ell+1}^{B'} \leftarrow \boldsymbol{W}_{\ell+1}^B \boldsymbol{P}^{\mathrm{T}}. \tag{4}$$

After applying all computed permutations to $\boldsymbol{\theta}_B$ to get $\boldsymbol{\theta}_{B'}$, the final merged model is computed as $\lambda\boldsymbol{\theta}_A + (1-\lambda)\boldsymbol{\theta}_{B'}$ for some $\lambda \in [0, 1]$.

## 3.2 Multi-Headed Attention

In finding corresponding neurons in the Multi-Headed Attention (MHA) parameters, namely the key ($\boldsymbol{W}_K$), query ($\boldsymbol{W}_Q$), value ($\boldsymbol{W}_V$), and linear layer ($\boldsymbol{W}_O$) weights, we propose several methods to compute potential permutation matrices.

For each of the key, query, and value weights, the full parameter $\boldsymbol{W} \in \mathbb{R}^{d_{\mathrm{model}} \times d_{\mathrm{model}}}$ is logically partitioned into $h$ attention heads each of output dimension $d_k = d_{\mathrm{model}}/h$ (Vaswani et al., 2017). In order to apply a permutation to these full weight matrices and maintain the functional equivalence of the overall model, permutations must operate on each attention head separately, and not permute features between attention heads. This is because the final hidden vector from MHA reflects a concatenation of the result from each head, which are computed separately with weights $\boldsymbol{W}_{K_i}, \boldsymbol{W}_{Q_i}, \boldsymbol{W}_{V_i}$ for head $i$.

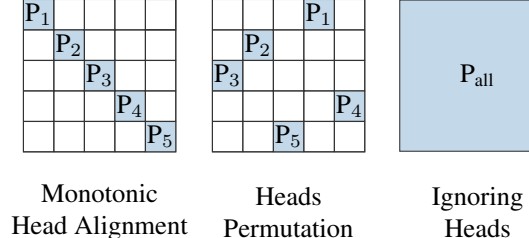

Monotonic Head Alignment     Heads Permutation     Ignoring Heads

Figure 2: Example permutation matrices resulting from different strategies for attention head alignment. Each $\boldsymbol{P}_i$ reflects permutations for features within attention heads.

Since our models are trained from distinct initializations, the correspondence of their attention heads may differ in addition to the correspondence of features within each head. This distinction is diagrammed in the first two permutations of Figure 2. We collect features from just after the attention computation and before the linear layer following attention. These features are the aforementioned concatenations. We first compute $\boldsymbol{C} = \mathrm{corr}(\boldsymbol{X}_A, \boldsymbol{X}_B)$ and then partition this correlation matrix by heads into $d_k \times d_k$ correlation matrices, for each potential attention head pair. Let $\boldsymbol{C}^{jk}$ be the block of the correlation matrix corresponding to head pair $(j, k)$. We then compute the optimal permutation for each unique head pair, and store its head-internal permutation and cost from the following:

$$\mathrm{cost}(j, k) = \max_\pi \sum_{i=1}^{d_k} \boldsymbol{C}_{i,\pi(i)}^{jk}. \tag{5}$$

We then compute the outer head correspondence permutation with a new assignment problem:

$$\pi_{\mathrm{outer}} = \arg\max_\pi \sum_{j=1}^{h} \mathrm{cost}(j, \pi(j)). \tag{6}$$

The outer permutation dictates the subset of previously computed inner permutations used in the final permutation, and the order in which to concatenate them, resulting in our 2-staged MHA permutation. We show this approach

---

**Algorithm 1** Multi-Headed Attention Permutation

---

    **Input:** Correlation Matrix $C$, number of heads, $h$
    **for** $i = 1$ **to** $h$ **do**
        **for** $j = i$ **to** $h$ **do**
            $\pi_{ij}, \text{costs}(i, j) = \text{LinearSumAssignment}(C^{ij})$
        **end for**
    **end for**
    $\pi_{\text{outer}} = \text{LinearSumAssignment}(\text{costs})$
    $\pi_{\text{final}} = \text{concat}(\pi_{i, \pi_{\text{outer}}(i)})$
    **Output:** $\pi_{\text{final}}$

---

in Algorithm 1. We note that LinearSumAssignment refers to the specific assignment problem we encounter in our method.

The resulting permutation matrix $P_{\text{MHA}}$ applies as following: $P_{\text{MHA}}^{\text{T}}$ can apply to $W_O$ as described previously, but we apply $P_{\text{MHA}}$ to each of $W_V, W_K$, and $W_Q$. We note that the $\langle W_{Q_i}x, W_{K_i}x \rangle$ dot-product attention operation multiplies away the head dimension $d_k$, meaning we do not necessarily have to use the same permutation for $W_V$ and $\{W_Q, W_K\}$. However, we still have to use the same outer permutation for all three matrices, as the attention weights resulting from $\langle W_{Q_i}x, W_{K_i}x \rangle$ are still head-specific. In experimentation, we do not find a notable difference between using a separate permutation for $\{W^Q, W^K\}^2$, and using the same permutation for $W_Q, W_K$ and $W_V$. Therefore, we consider only the latter case for simplicity.

We show an example of an output from our proposed algorithm in Figure 2, as well as some alternative approaches to permuting MHA weights. The first permutation matrix shows the resulting block-diagonal structure of assuming that heads are aligned between different minima. The second matrix shows an example from our method, and the third matrix disregards head structure and allows permuting features across heads. We note that ignoring heads will not lead to a valid permutation $\pi$ where $f(x; \theta) = f(x; \pi(\theta))$, but we still include it for experimental comparisons.

## 3.3 Residual Connections

Each Transformer layer, assuming no cross-attention, contains two residual connections. This means all layers, from the embedding layer to the output layer, is linked via consecutive residual connections. This diverges even from ResNets, which generally contain skip-connections every 2 or 3 layers (He et al., 2016).

We diagram the residual connections of a Transformer layer and their relationships to model parameters in Figure 1. The first connection skips past the multi-headed attention sublayer, and the second connection skips past the feed-forward sublayers, as diagrammed. The connections can be formulated as the following, where LN refers to LayerNorm:

$$
\begin{aligned}
x_a^r &= \text{LN}(W_O \text{MHA}(x) + x), \\
x_f^r &= \text{LN}(W_2 \text{ReLU}(W_1 x_a^r) + x_a^r).
\end{aligned}
\tag{7}
$$

As seen in the equations, the input and output of both sublayers are added to create a new output, which implies that if a permutation operation is applied to the output state, the permutation needs to be the same for both addends.

We note that the addends are normalized via the LayerNorm module, and any permutation to the output would need to permute the features of the LayerNorm module as well. We apply the permutation to the weights of LN. Because LN is not a full weight matrix, and maintains the same feature ordering as its input, we must apply the permutation to the addends of the residual connections as well (Stoica et al., 2024).

---

[2]This permutation would be computed from features $W_Q x$ and $W_K x$

Now, ignoring the parameters from the LayerNorm module for a moment, we see that applying the permutation to the output of the second residual connection leads to the following:

$$\begin{aligned}
\boldsymbol{P}\boldsymbol{x}_f^r &= \boldsymbol{P}\left(\boldsymbol{W}_2\text{ReLU}(\boldsymbol{W}_1\boldsymbol{x}_a^r) + \boldsymbol{x}_a^r\right) \\
&= \boldsymbol{P}\boldsymbol{W}_2\text{ReLU}(\boldsymbol{W}_1\boldsymbol{x}_a^r) + \boldsymbol{P}\boldsymbol{x}_a^r \\
&= \boldsymbol{P}\boldsymbol{W}_2\text{ReLU}(\boldsymbol{W}_1\boldsymbol{x}_a^r) + \boldsymbol{P}(\boldsymbol{W}_O\text{MHA}(\boldsymbol{x}) + \boldsymbol{x}).
\end{aligned} \tag{8}$$

We see that due to the residual structure, any permutation applied to second feed-forward weight parameter, $\boldsymbol{W}_2$, must also be applied to MHA, or more specifically the $\boldsymbol{W}_O$ matrix. To unpermute these features, we apply $\boldsymbol{P}^{\mathrm{T}}$ to where the permuted $\boldsymbol{x}_a^r$ and $\boldsymbol{x}_f^r$ states become inputs, which are $\boldsymbol{W}_1$ and $\{\boldsymbol{W}_Q, \boldsymbol{W}_V, \boldsymbol{W}_K\}$, respectively.

Because the input to each layer must be permuted $\boldsymbol{P}\boldsymbol{x}$, and the output of each layer is also permuted $\boldsymbol{P}\boldsymbol{x}_f^r$, we can see that the entire Transformer architecture uses the same $\{\boldsymbol{P}, \boldsymbol{P}^{\mathrm{T}}\}$ matrices for all the weights involved in residual connections. This is unlike our other proposed permutations for multi-headed attention which are specific to each Transformer layer. Requiring the same permutation throughout the model reduces the degrees of freedom available in finding an optimal permutation, which is an important result demonstrating a potential difficulty of aligning Transformers.

At the ends of the models, namely the embedding layer(s) and output layer(s), we also apply these transformations, as the input to the first Transformer block and the output of the last Transformer block are permuted. We apply this $\boldsymbol{P}$ to the embedding weights, including positional and any special token embeddings, and at the final layer, we apply $\boldsymbol{P}^{\mathrm{T}}$ to the weight matrix immediately following the last Transformer block LayerNorm. This is usually a pooling or dense layer, depending on the model task.

Because of the multiple potential features that could contribute to the computation of the residual permutations, namely both $\boldsymbol{x}_a^r$ and $\boldsymbol{x}_f^r$ across all layers, we consider several strategies for learning these mappings. We test 4 approaches: *First* refers to only obtaining features from the features immediately following the embedding layer(s). *Last* refers to only obtaining features from just after the final Transformer layer's LayerNorm. *All* refers to concatenating all features $\boldsymbol{x}_f^r$, and $\boldsymbol{x}_a^r$ from all Transformer layers, and *separate* refers to computing 2 $\{\boldsymbol{P}, \boldsymbol{P}^{\mathrm{T}}\}$ pairs per Transformer layer, one for $\boldsymbol{x}_f^r$, and the other for $\boldsymbol{x}_a^r$. The *separate* approach also does not lead to a valid permutation like *Ignore Heads* in Section 3.2, but we also include it for experimental comparisons.

### 3.4 Feed-Forward and Output Layers

Unlike the residual stream or Multi-Headed attention, Feed-Forward sub-layers require no special attention in order to permute them into a new space. We simply compute correlations from the features after the computation of the first Feed-Forward layer ($\boldsymbol{W}_1$), and compute $\boldsymbol{P}, \boldsymbol{P}^{\mathrm{T}}$ separately for each Transformer layer. We apply these permutations as described in Section 3.1:

$$\begin{aligned}
\boldsymbol{W}_1' &\leftarrow \boldsymbol{P}\boldsymbol{W}_1 \\
\boldsymbol{W}_2' &\leftarrow \boldsymbol{W}_2\boldsymbol{P}^{\mathrm{T}}
\end{aligned} \tag{9}$$

As many models have task-specific output layers like classification heads or masked-language modeling heads, there is also another permutation able to be computed between a pooling/dense layer and the actual model head, which tends to also be a linear layer. This permutation mapping would proceed as normal, like the feed-forward example, but we do not include it in our experimentation as we see no notable differences in its inclusion.

## 4 Experimental Settings

### 4.1 Models

We investigate Transformer encoder-based masked language models in this work. Specifically, we consider 5 different BERT models, seeds 1 through 5, from the MultiBERTs reproductions (Devlin et al., 2019; Sellam et al., 2021). Each of these models is a `bert-base-uncased` checkpoint, but trained with a different random initialization and random batch ordering. These properties are the type of SGD-related variation we seek to study between different minima. All

models use the same original BERT vocabulary and tokenizer. All of our reported experiments include a mean and standard error across the 10 unique pairings resulting from our 5 different models.

For classification tasks, we fine-tune each of the MultiBERTs models with a randomly initialized classification head, including pooling layer and classification layer weights. We keep the head initializations the same across models.

We report vanilla averaging as our main baseline for comparison, computed as $\boldsymbol{\theta}_{\mathrm{avg}} = \frac{1}{2}(\boldsymbol{\theta}_A + \boldsymbol{\theta}_B)$.

### 4.2  Tasks and Datasets

We focus on two different tasks to test our method. We use the masked language modeling task to test our base method, as this is the main task that BERT and many other pretrained Transformer encoded models are trained with. We also consider fine-tuning these models for classification tasks, and then comparing them in their fine-tuned state. We use the General Language Understaning Evaluation (GLUE) benchmark (Wang et al., 2018) for our classification tasks, and exclude WNLI as in Devlin et al. (2019). GLUE is a set of 8 diverse natural language understanding classification tasks. We report scores on GLUE test sets from our reproductions in Table 4 in Appendix A.

For our experiments on masked language modeling, we use the validation set of the Wikitext-103 benchmark as our evaluation data Merity et al. (2016)[3]. For computing model activations, we extract a random sample of just over 1 million sentences of the Books corpus Zhu et al. (2015). For a majority of our experiments, we sample 100k sentences and use this subset to compute features, unless stated otherwise. We take a diverse sample across different genres among the books available. We choose the Books corpus as it is part of the original pre-training data from BERT (Devlin et al., 2019).

For GLUE experiments, we use the full training data for each of the tasks to compute features, and the full validation sets to compute losses. The amount of data available for each task varies, and statistics are also reported in Table 4 in Appendix A.

### 4.3  Evaluation

To compute loss barriers, we compute several interpolations of $\boldsymbol{\theta}_A$ and $\boldsymbol{\theta}_B$, as $\lambda\boldsymbol{\theta}_A + (1 - \lambda)\boldsymbol{\theta}_B$. Specifically, we use 21 samples evenly spaced between $\lambda = 0$ and $\lambda = 1$, inclusive. We use the definition of loss-barrier as Frankle & Carbin (2018)[4], defined as the maximum difference between the loss of an interpolation and the average loss of the base models:

$$\max_{\lambda} \mathcal{L}(\lambda\boldsymbol{\theta}_A + (1 - \lambda)\boldsymbol{\theta}_B) - \frac{1}{2}(\mathcal{L}(\boldsymbol{\theta}_A) + \mathcal{L}(\boldsymbol{\theta}_B)). \tag{10}$$

To compute MLM loss/pseudo-perplexity, we use a masking probability of $p = 0.15$ across block sizes of 128 tokens. For $N$ masked samples in text $\mathbf{W}$, we compute pseudo-perplexity as:

$$\mathrm{Pseudo\text{-}PPL}(\mathbf{W}; \boldsymbol{\theta}) = 2^{-\frac{1}{N}\sum_{i=1}^{N} \log_2 p_{\boldsymbol{\theta}}(w_i|\mathbf{W}_{-i})}. \tag{11}$$

We report loss on GLUE tasks as normal, defined by each task, across the entire validation set.

## 5  Results and Analysis

### 5.1  By component

We report results on the 10 MultiBERTs merges after merging different sets of Transformer components described in Section 3. We show pseudo-perplexity results across the range of interpolations, displayed Figure 3. We report vanilla averaging with no permutations, merging all feed-forward sublayers, merging all multi-headed attention sublayers, and merging all feed-forward sublayers and all multi-headed attention sublayers. Aligning and merging either the feed-forward sublayers or the attention sublayers clearly leads to a perplexity reduction over the baseline, and their combination leads to a stark reduction, of almost $7\times$ the original perplexity at $\lambda = 0.5$. We do not include permuting

---

[3]We obtain the `wikitext-103-raw-v1` version, available from `https://huggingface.co/datasets/wikitext`
[4]Referred to as *linear interpolation instability* in this work.

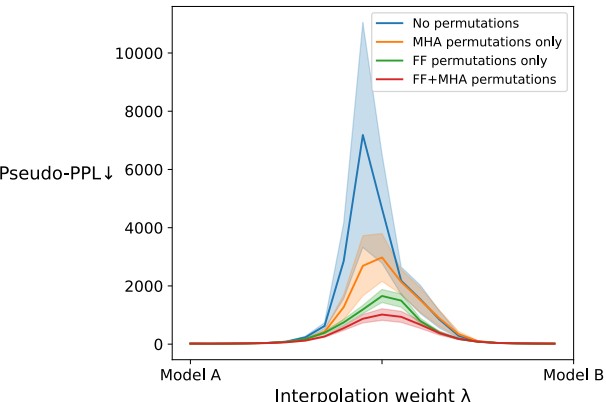

Figure 3: Pseudo-perplexity scores of BERTs, trained on the masked language modeling task, combined using our method. Curves differ by which components they merge. Results across 10 merges are shown with standard error regions shaded around each curve. Each additional merged component leads to further barrier reduction.

parameters involved in residual connections (Section 3.3) and the output projection (Section 3.4) here as they do not outperform merging only feed-forward and attention sublayers. We discuss the residual connections further in Section 5.3.

The consistently reduced barrier between minima indicates that these different models are connected with a lower loss path than seen without considering these models within a more similar loss space. We note that we do not observe a linear or convex loss path between these models, as sometimes observed in previous work on MLPs, ResNets, and VGGs (Tatro et al., 2020; Entezari et al., 2022; Ainsworth et al., 2023). In this line of prior work, the same data is generally used to train models, compute activations for alignment and merging, and compute loss barriers. Due to the extensive pretraining data of these masked language models, and the limited alignment and evaluation data we use, we do not test for linear mode connectivity in the same manner. Instead, the stark loss reduction seems to indicate that minima are connected with a barrier *at least as high* as what we report.

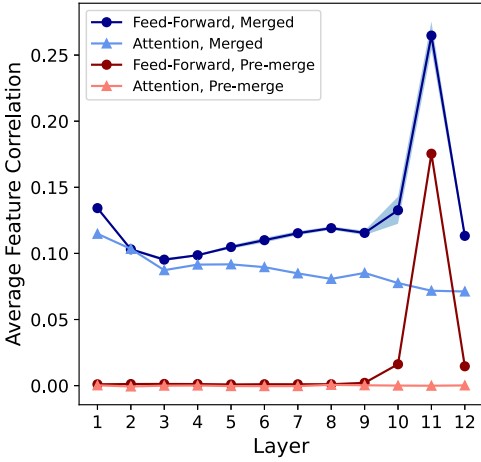

Figure 4: Average feature correlations between layers from different MultiBERTs. We report correlations for both Feed-Forward and MHA features. Both components see much higher average correlation after applying their respective component permutations. Values are averaged over 10 merges, with standard error regions shaded.

In understanding the extent to which these different Transformers learn similar representations throughout the model, we compute the average feature correlations between all 10 masked language model pairs. We report correlations for attention and feed-forward layer features before and after applying our method. Individual correlations are computed

between features of the same index. We show these values in Figure 4. We find that the average feature correlations of aligned models are significantly higher than those of the original models, which demonstrates some success of our alignment procedure, but still no higher than 0.3. Previous work has reported higher average correlations on different architectures like ResNets (Tatro et al., 2020), but in the image domain. Some pre-trained transformers are also known to be sparsely activated and able to be pruned heavily (Li et al., 2023; Dalvi et al., 2020), which may also lead to lower average feature correlations.

## 5.2 Multi-headed attention

Table 1: Loss Barriers of merged MultiBERTs with feed-forward and attention components merged. Methods describe which MHA algorithm was applied. Loss barriers are the largest difference between an interpolation and the average of the individual model losses. Maintaining head structure in the permutation while allowing different head correspondences between models is the most optimal permutation.

| Method | Loss Barrier ↓ | Std. Err. |
|---|---|---|
| *Vanilla Attention Avg.* | *4.31* | *0.21* |
| Monotonic Head Alignment | 4.13 | 0.20 |
| Ignore-Heads | 3.97 | 0.25 |
| Head-Perm | **3.71** | 0.23 |

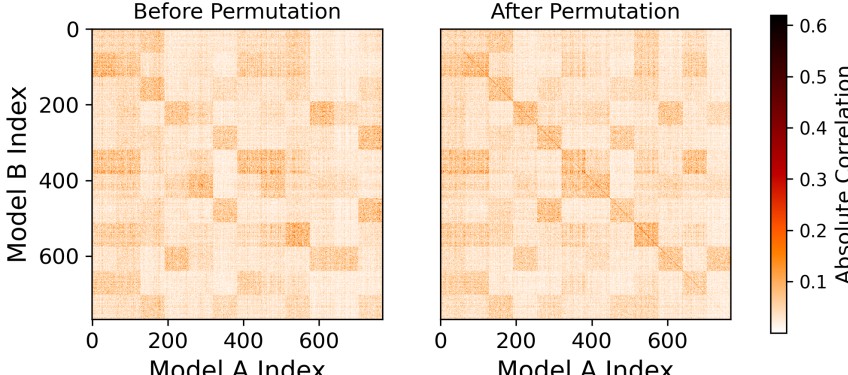

Figure 5: Visualization of correlation matrices between features before and after permuting. These features are from the seventh multi-headed attention layer from 2 different MultiBERTs models. On the left, 12 attention head boundaries are clearly visible, and highly correlated regions do not necessarily correspond to the same attention head indices, supporting our two-stage permutation method. On the right, the two-stage permutation method outcome can be seen via the dark diagonal line, and its surrounding block diagonal pattern.

We report loss barriers for our Head-Permutation multi-headed attention approach as compared to some alternatives also described in Section 3.2 in Table 1. These results reflect permuting both attention parameters as well as feed-forward parameters. We see that our proposed Head-Permutation approach for the attention sub-layer outperforms simple attention averaging, as well as approaches ignoring the multi-headed structure of the weight parameters (*Ignore-Heads*), and not allowing for different head correspondences across different models (*Monotonic*). We also show an example correlation matrix between the first multi-headed attention layer from 2 different MultiBERTs models in Figure 5. The correlation matrix shows clear attention head boundaries, as well as a scattered pattern that supports our proposed technique that does not assume any monotonically ordered head correspondence.

## 5.3 Residual Stream

As described in Section 3.3, many of the permutation operations in the Transformer architecture are shared due to its repeated Add&Norm components. This linkage results in a huge reduction in the number of permutation symmetries of

the Transformer architecture, and therefore the number of valid permutations of parameters involved in the residual stream. We report the loss barriers after applying our permutation alignment to only the parameters involved in the residual connections in Table 2. We find that an identity permutation performs significantly better than all other proposed approaches. Even the separate permutation approach does not outperform using the identity matrix $I_d$ as the residual merge. This approach does not even create a valid Transformer symmetry, but has many more degrees of freedom than the other proposed approaches.

Table 2: Loss Barriers of merged MultiBERTs with only residual components merged. Methods describe which features were used to compute permutations. Loss barriers are the largest difference between an interpolation and the average of the individal model losses. As there is only one $\{P, P^T\}$ pair for the entire residual stream, identity permutations outperform our learned approaches.

| Method | Loss Barrier $\downarrow$ | Std. Err. |
|---|---|---|
| Identity | **4.95** | 0.38 |
| First | 7.58 | 0.19 |
| Last | 7.41 | 0.18 |
| All | 7.34 | 0.22 |
| Separate | 9.38 | 0.49 |

## 5.4 Amount of Data

We report loss barriers on the masked language modeling task while varying the amount of sentences used to compute correlations. All previous experiments are reported on $\sim$100k sentences. We combine feed-forward and attention layers in this setting. Results are shown in Figure 6. Values are fairly similar across data amounts, suggesting that there is no strong relationship between the amount of data used and the overall loss barrier. We do see, however, small variations between different data amounts, which are likely due to the content of the different sets of sentences used to create the correlation matrices. We believe that more than the quantity of tokens, the type and quality of text used for computing correlations likely matters more. We leave further investigation into specific data sources for aligning text-based Transformer models for future work.

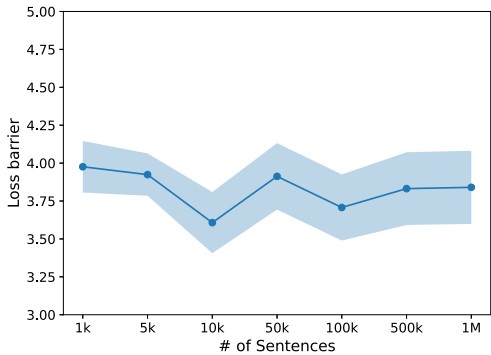

Figure 6: Loss barriers on the masked language modeling task with different data amounts used for computing features. Standard error regions are shaded. The amount of data does not have a strong directional relationship with the size of the loss barrier in this setting.

## 5.5 GLUE Results

We investigate loss barriers between fine-tuned BERT models across 8 different GLUE tasks. We compare loss barriers from vanilla averaging to those obtained after applying our method to combine the models. Different from our result on masked language modeling, we include residual permutations as we find it has a slight decrease in loss barriers over

just feed-forward and attention permutations. We use the *First* approach to compute residual permutations. We report these results in Table 3. We additionally include loss barrier curves for all 8 tasks in Appendix B in Figure 7.

Table 3: Loss barriers for both vanilla averaging of fine-tuned BERT models, and barriers for BERT models merged with our method. In this setting, we merge feed-forward, attention, and residual components. Residual mergers are conducted using the *First* strategy. 6 of 8 tasks see a loss barrier reduction in using our method.

| | Vanilla averaging | | Ours | |
|---|---|---|---|---|
| | Barrier ↓ | Error | Barrier ↓ | Error |
| MNLI-mm | **0.61** | 0.03 | 0.72 | 0.08 |
| QQP | 1.37 | 0.09 | **1.20** | 0.11 |
| QNLI | **0.64** | 0.04 | 0.77 | 0.06 |
| SST-2 | 0.42 | 0.04 | **0.36** | 0.07 |
| CoLA | 1.31 | 0.14 | **1.11** | 0.13 |
| STS-B | 5.15 | 0.44 | **4.24** | 0.35 |
| MRPC | 2.74 | 0.08 | **1.93** | 0.11 |
| RTE | 0.53 | 0.04 | **0.41** | 0.05 |

While we are able to observe lower loss barriers between minima, the trends of loss reduction across interpolations is inconsistent, especially compared to the masked language modeling setting. For example, while the maximum loss of vanilla averaging is higher than the maximum loss of our approach, there are still other interpolation weights where the loss of our approach is instead higher than vanilla averaging. We also note an interesting pattern of lower loss than either parent model for several tasks around $\lambda = 0.15$ and again at $\lambda = 0.85$. Additionally, the loss barrier curves of some vanilla merges between these fine-tuned models have different behavior than their pretrained alternatives, as seen in Figure 7 with the presence of "M" like loss curve shapes between minima. While our method can extend to this setting to find a lower loss path between fine-tuned minima, it is unclear which kind of data is necessary to observe the lowest loss path. We leave further investigation into connecting fine-tuned models, and further understanding of their pre-merging connectivity in loss space, for future work.

## 6 Discussion and Conclusion

By considering the set of functionally equivalent Transformers reachable using permutation mappings, we can consistently find linear paths between models with lower loss than the path obtained via vanilla interpolation. This conclusion about the connectedness between these models has implications on our understanding of the "smoothness" of Transformer loss space and the sharpness of their minima; in comparing minima using our method, they are far less isolated than previously understood. This understanding of the geometric properties of minima can have implications in how we design optimization methods, ensembles of models, and additional merging techniques. For example, it is widely contested whether sharp minima can generalize as well as flat minima across many deep learning models Keskar et al. (2016); Dinh et al. (2017). In our work, we show that it is necessary to consider permutation invariances of Transformer models when characterizing the geometric properties of their minima.

As we take only a first attempt at connecting separately trained Transformers along a lower loss path, there is much room for future work in understanding Transformer loss landscapes. A deeper understanding of the relationships between fine-tuned models, Transformer width and loss barriers, and the data needed to compute more informative correlations is needed in order to further characterize the relationship between Transformer minima.

**Broader Impact Statement**

While it is already difficult to meaningfully characterize the capabilities of a trained model, it is even more difficult to know what an interpolated model's capabilities are, with respect to their parent models. Our work focuses on the losses of these interpolated models rather of their performance in other aspects. Outside of loss, it is unclear how the performance of these models changes when combined. Thorough testing of interpolated models should be investigated before actual use.

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

## A    GLUE Reproduction

We report the average values of our GLUE reproductions across the MultiBERTs models here. We train MNLI-mismatched, QQP, QNLI, SST-2, CoLA, STS-B, and RTE tasks for 3 epochs, and we train MRPC for 5 epochs. We follow all other hyperparameters of the reproduction implemented in Ren et al. (2023).

Table 4: Results on GLUE for both the original BERT model Devlin et al. (2019), and our reproduction across MultiBERTs models 1-5. We report the average and the standard deviation across 5 models. Accuracy is reported for MNLI-mm, QNLI, SST-2, and RTE. F1 scores are reported for QQP and MNLI. Matthews correleation is reported for CoLA, and Spearman-r correlation is reported for STS-B.

| Task | MNLI-mm | QQP | QNLI | SST-2 | CoLA | STS-B | MRPC | RTE |
|---|---|---|---|---|---|---|---|---|
| **Training instances** | 392k | 363k | 108k | 67k | 8.5k | 5.7k | 3.7k | 2.5k |
| **Validation instances** | 9.8k | 40.4k | 5.5k | 0.9k | 1k | 1.5k | 0.4k | 0.3k |
| BERT$_{\text{BASE}}$ | 83.4 | 71.2 | 90.5 | 93.5 | 52.1 | 85.8 | 88.9 | 66.4 |
| Our Reproduction | $84.3_{\pm0.002}$ | $87.3_{\pm0.001}$ | $91.0_{\pm0.003}$ | $91.7_{\pm0.002}$ | $58.3_{\pm0.011}$ | $89.2_{\pm0.003}$ | $90.7_{\pm0.010}$ | $66.2_{\pm0.063}$ |

## B    GLUE Loss Plots

We plot loss barriers for vanilla merging and our method across 8 GLUE tasks. For a majority of the tasks, the loss barrier of interpolations is smaller than that of vanilla merging.

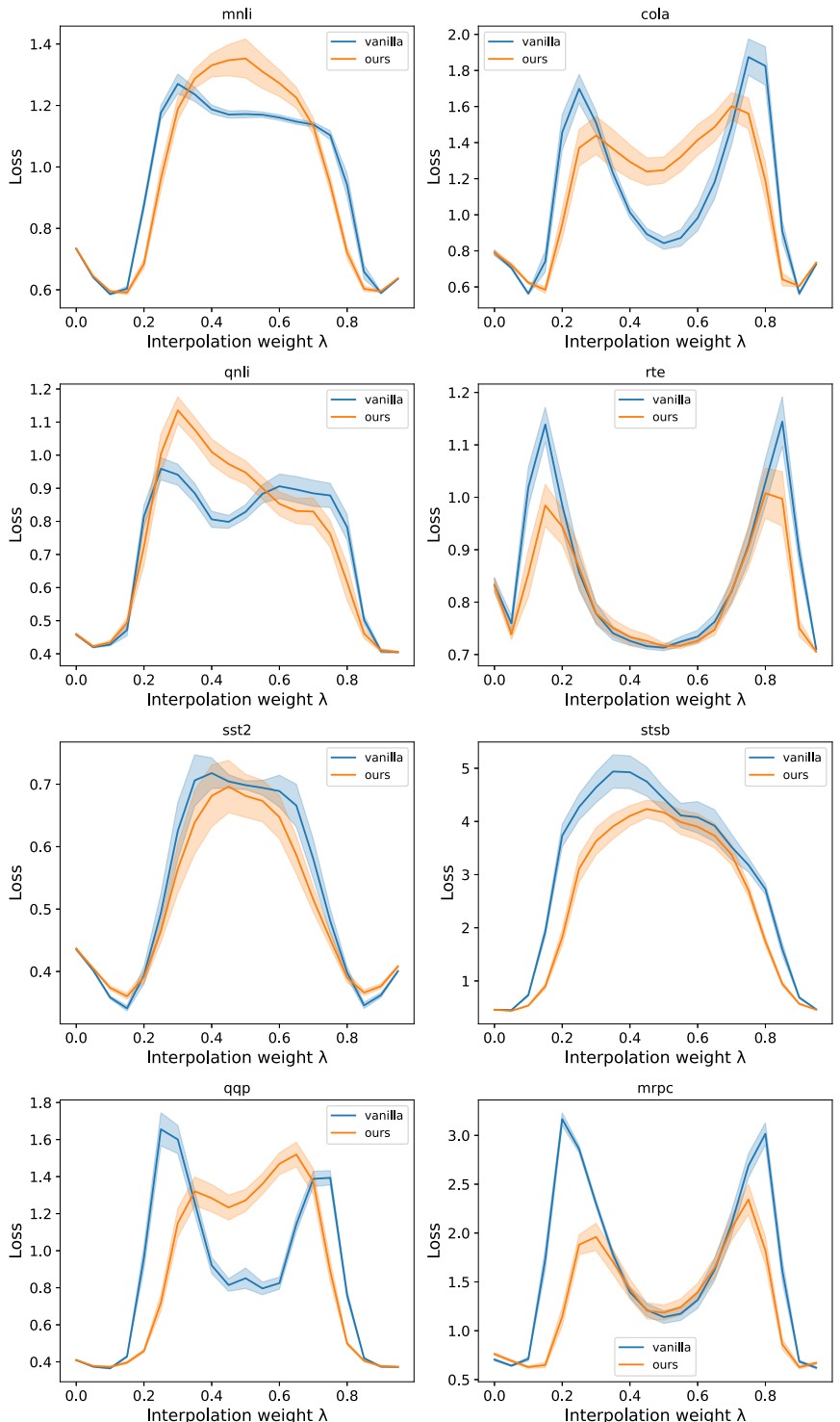

Figure 7: Loss barriers on 8 GLUE tasks for both the vanilla interpolation and our merging strategy.

