# OpenReview forum: "Merging Text Transformer Models from Different Initializations"
_TMLR — Accepted by TMLR_

### Review · Reviewer_3kxD · 2024-07-05

**Summary Of Contributions:**

This paper investigates the merging of separately trained transformer models to understand their loss landscape connectivity. It introduces a permutation-based model merging algorithm tailored for transformers, addressing specific architectural challenges like multi-head attention and residual connections. The results show that this method consistently reduces loss barriers between models, indicating that transformer minima are less isolated than previously thought, and suggesting that similar representations are learned even from different initializations.

**Audience:**

Yes

**Broader Impact Concerns:**

Model merging is an important aspect when looking at effective distributed learning - especially for LLMs (which are currently mostly transformer-based). From that perspective, the contribution might be very useful in practice. However, it should be clarified better, why it fails in some cases and whether these cases could be identified on-the-fly without checking plain averaging for comparison.

**Claims And Evidence:**

Yes

**Requested Changes:**

The authors should add more details on the permutation operation. It would make the paper more accessible to know how the "optimal" permutation is found.

Figure 3 should be more self-explaining.

**Strengths And Weaknesses:**

**Strengths**

The paper is overall well written and easy to follow. The methods part is described precisely enough to understand the merging procedure properly.

The proposed method is simple and effective. The results (at least partially) underline that.

**Weaknesses**

To me it appears unclear why in two out of eight cases (Table 3) the presented permutation-based merging performs worse than vanilla averaging. I think that this should be addressed in more detail. What happens in these cases? How do the correlations look like? Is it a failure of the permutation search?

One debatable point of criticism of the paper is its incremental nature. The fundamental methodological framework is based on Ainsworth et al. (2023) (cf. also Yamada 2023). In the present work, however, this method has been adapted to work with transformers in a somewhat straightforward manner. Although it is very reasonable how this has been accomplished, it should be discussed whether this addition brings enough innovative novelty for publication in TMLR.

Masanori Yamada, Tomoya Yamashita, Shin'ya Yamaguchi, Daiki Chijiwa. Revisiting Permutation Symmetry for Merging Models between Different Datasets. 2023. https://arxiv.org/pdf/2306.05641

---

> ### Author Response · Authors · 2024-08-11
> **Author Response to reviewer 3kxD**
>
> Thank you for your time, valuable feedback, and engagement in your review. We first describe our steps to address the requested changes, and then discuss the questions and comments from the weaknesses listed.
>
> **Requested changes**
>
> 1. Requested change 1: More detail on the permutation operation.
>
> We have added more detail on the optimization problem and permutation solution throughout section 3.1 in the revised PDF.
>
>
> 2. Requested change 2: Figure 3 should be more self-explaining
>
> We have adjusted Figure 3 in the revised PDF to be more descriptive in the legend labels, changed the x-axis to be more clear about what is being interpolated, and changed the y-axis to show that lower Pseudo-PPL is better. Thanks for helping us improve our clarity.
>
> **Discussion of weaknesses**
>
> 1. Weakness 1: What could cause the 2 negative results in table 3?
>
> Merging fine-tuned models from different initializations poses a unique challenge of how best to align them given their 2-stage training procedure. This has not yet been addressed in previous mode connectivity literature (to the best of our knowledge).
> QNLI and MNLI have large training data sizes compared to the rest of GLUE (Appendix A). It may be the case that due to longer fine-tuning, the models become more difficult to merge in loss space. It is also unclear which data should be used to align these models.
>
>
> 2. Weakness 2: Incremental Nature
>
> While it is true that the framework of the paper builds off of a line of prior work like Ainsworth et al. 2023 (also Yamada et al. 2023), [Tatro et al. 2020](https://proceedings.neurips.cc/paper/2020/file/aecad42329922dfc97eee948606e1f8e-Paper.pdf), and [Li et al. 2016](https://arxiv.org/pdf/1511.07543), we feel that extending this type of work to Transformers is not a straightforward pursuit. We highlight the unique contributions of this work:
>
> a. A thorough description of permutation invariances in the Transformer architecture that preserve functional equivalence
>
> b. Multi-headed attention merging algorithm with ablation studies, requiring the consideration of splitting, dot-product attention, and concatenating separate head results.
>
> c. A discussion of permuting features involved in residual connections, and results suggesting this may be a unique difficulty in aligning transformers (even more than ResNets). More information about this comparison has been added to section 3.3.
>
> d. Exploring the alignment of separately pre-trained *and* subsequently fine-tuned models
>
> e. Applying these methods to and evaluating in the text domain

---

### Review · Reviewer_S6LB · 2024-07-17

**Summary Of Contributions:**

This paper proposes a permutation-based merging method for transformers and give analyses on the loss barrier in the setting of linear mode connectivity. The merging method considers the specific structure of transformers such as multi-head attention and outperforms the vanilla interpolation method and naive interpolation methods that do not consider the structure. The experimental results provide valuable insights on the landscape of the loss values of transformers e.g. the minima may be far less isolated than previously understood.

**Audience:**

Yes

**Broader Impact Concerns:**

No concerns.

**Claims And Evidence:**

Yes

**Requested Changes:**

I would love to see comparisons with existing methods, at least ones mentioned in the paper if appropriate.

It would be good to see finetuning or other experiments that show better performance for the merged model.

It would be good to have discussions on why model merging has been challenging for transformers. The model is of course more complicated than MLPs and CNNs, but the components are basically matrix multiplications and it does not seem that it is "theoretically" more challenging.

**Strengths And Weaknesses:**

Strengths:

It provides a simple and intuitive way to merge transformer weights. The experimental results show that it provides lower loss barrier values compared to simpler merging methods.

It explains the method in detail. It will help readers to reproduce the results.

Weaknesses:

There is no comparison with prior studies merging transformers e.g. Imfeld et al. (2023). The conditions may not be exactly the same, but still the method should be still able to be used for comparisons.

It is not shown that the merged model can be actually useful. I think it is a good empirical result that the loss barrier can be lowered by tailored permutation-based merging methods, but if the merged model is not actually useful, the value of such merging is questionable. Imfeld et al. (2023) shows the merged model can be better after finetuning.

It is a natural extension of existing permutation-based methods to transformers and the results are not surprising and may be less novel.

It does not have theoretical discussions. Why are transformers considered as special?

---

> ### Author Response · Authors · 2024-08-11
> **Author response to reviewer S6LB**
>
> Thank you for your time, valuable feedback, and engagement in your review. We first describe our steps to address the requested changes, which also serves as our discussion of the weaknesses since the two are heavily linked in the review.
>
>
> 1. Requested change/weakness on why model merging is challenging for Transformers
>
> In the revised PDF, we include additional details in the discussion section about what makes Transformers uniquely difficult. We point to the discussion of MHA features at the end of paragraph 2 in section 3.2 which addresses dealing with MHA concatenation step, and section 3.3 where we added information about how the residual structure of Transformers is even denser than ResNets and this makes finding an ideal permutation even more difficult. These structures go beyond basic MLPs (matrix multiplications), and require more intervention than what is described initially in section 3.1.
>
> 2. Requested change on fine-tuning experiments & comparisons
>
> We focus on using model merging as a tool to study relationships between minima in this work. This requires preserving the functional equivalence of the model $f(x; \theta)$ after permutations $\pi$ such that $f(x; \theta) = f(x; \pi(\theta))$. To the best of our knowledge, there are not other techniques that 1) apply to Transformers straightforwardly and 2) preserve equivalence allowing us to study these minima relationships. We are aware of the method from Imfeld et al. (2023), and do not compare for this reason.

---

### Review · Reviewer_MkbX · 2024-08-01

**Summary Of Contributions:**

This paper investigates the loss of landscape connectivity between separately trained (different initializations) Transformer models. The authors propose a model merging technique that aligns representations from separate models using permutation-based methods for different components of the Transformer architecture, including multi-headed attention, residual connections and feed-forward layers. The main merging technique is as follows; compute correlation matrices between features of the two models for different layers and components, and find optimal permutations that align the features using the Jonker-Volgenant algorithm, then apply these permutations to specific components (e.g., multi-head attention) of one model to align it with the other. The authors demonstrate that their method consistently finds lower loss barriers between model minima compared to simple model averaging, both for pre-trained masked language models and models fine-tuned on GLUE tasks. The results suggest that Transformer minima are less isolated in the loss landscape than previously understood when considering permutation symmetries. In their experiments, the authors show that merging both feed-forward and attention sublayers leads to a significant reduction in perplexity, almost 7 times lower than the original. They also find that the amount of data used to compute correlations for alignment does not strongly affect the loss barrier. For fine-tuned models on GLUE tasks, the method reduces loss barriers in 6 out of 8 tasks compared to vanilla averaging, though the trends are less consistent than in the masked language modeling setting.

**Audience:**

Yes

**Claims And Evidence:**

Yes

**Requested Changes:**

N/A

**Strengths And Weaknesses:**

**Strengths:**

- Novel contribution in extending permutation-based model merging techniques to Transformer, addressing a 2-staged approach for multi-headed attention.
- Comprehensive empirical evaluation across multiple settings, including masked language modeling and fine-tuned models on GLUE tasks.
- Provides insights into the geometry of Transformer loss landscapes, suggesting implications for optimization methods and ensemble techniques.
- Clear methodology and detailed ablation studies examine different components of the proposed approach.

**Weaknesses:**

- Limited theoretical analysis or guarantees for the proposed method, relying primarily on empirical results – why post-activation features are used?
- The approach does not consistently outperform vanilla averaging for all GLUE tasks, and the reasons for this variability are not fully explored.
- The paper does not extensively compare the proposed method to other recent model merging techniques beyond vanilla averaging.
- While lower loss barriers are demonstrated, the practical implications or benefits of this finding (e.g., for downstream tasks or model compression) are not thoroughly explored.

---

> ### Author Response · Authors · 2024-08-11
> **Author Response to reviewer MkbX**
>
> Thank you for your time, valuable feedback, and engagement in your review. We discuss the questions and comments from the weaknesses listed.
>
> 1. Why are post-activation features used?
>
> This detail only applies to feed-forward sublayer features, referring to post-GELU features. The only other activation present in the model layer is the attention softmax, which is applied before multiplying with value vectors. We collect our attention features just after the value vector multiplication, so the activation is not relevant in this case. We use post-activation as we anecdotally found it more stable to align during experimentation, and in line with prior work which found it to work better than pre-activations (Tatro et al. 2020)
>
> 2. GLUE results are inconsistent and reasons are not entirely explored
>
> Merging fine-tuned models from different initializations poses a unique challenge of how best to align them given their 2-stage training procedure. This has not yet been addressed in previous mode connectivity literature (to the best of our knowledge).
> QNLI and MNLI have large training data sizes compared to the rest of GLUE (Appendix A). It may be the case that due to longer fine-tuning, the models become more difficult to merge in loss space. It is also unclear which data should be used to align these models.
>
> 3. Other model merging comparisons
>
> We focus on using model merging as a tool to study relationships between Transformer minima in this work. This requires preserving the functional equivalence of the model $f(x; \theta)$ after permuting with $\pi$ such that $f(x; \theta) = f(x; \pi(\theta))$. To the best of our knowledge, there are not other techniques that 1) apply to Transformers straightforwardly and 2) preserve this equivalence allowing us to study these minima relationships.
>
> 4. Practical applications
>
> We acknowledge that the lack of immediate applicability is a limitation of our work. The goal of our work is to better understand the relationship between Transformer SGD solutions in the text setting. This understanding may have an impact in downstream practical applications (minimization techniques that consider local geometry, training dynamics, model selection), but is not the focus of our work.

---

### Decision · Action_Editor_kbRf · 2024-10-08

**Recommendation:** Accept as is

**Comment:**

This paper explores the merging of separately trained transformer models to analyze their loss landscape connectivity. It introduces a permutation-based merging method that accounts for transformer-specific structures like multi-head attention and residual connections. The method outperforms standard interpolation techniques by reducing the loss barriers between models, providing evidence that transformer minima are less isolated than previously believed. The results suggest that despite different initializations, transformers learn similar representations, offering new insights into the structure and connectivity of their loss landscapes.

All reviewers vote for acceptance, though they raised some concerns. However, the concerns are primarily about the novelty of the method. As per TMLR's policy, novelty alone cannot be grounds for rejection, and the method is well evaluated. Additionally, the authors responded to the reviewers' comments effectively. Therefore, the paper can be accepted by TMLR.

**Audience:**

Model merging is currently a hot topic in machine learning. Thus, the paper can attract ML researchers.

**Claims And Evidence:**

The experimental results support the claims.